# Evolution of Rail Contact Fatigue on Crossing Nose Rail Based on Long Short-Term Memory

**Lei Kou** [1] , **Mykola Sysyn** [1,\*] , **Jianxing Liu** [2], **Olga Nabochenko** [1], **Yue Han** [3], **Dai Peng** [4] **and Szabolcs Fischer** [5,\*]

1 Institute of Railway Systems and Public Transport, Technical University Dresden, 01069 Dresden, Germany
2 School of Civil Engineering, Southwest Jiaotong University, Chengdu 610031, China
3 China Railway Hohhot Group Co., Ltd., Hohhot 012000, China
4 Infrastructure Inspection Research Institute, China Academy of Railway Sciences, Beijing 100081, China
5 Department of Transport Infrastructure and Water Resources Engineering, Faculty of Architecture, Civil and Transport Engineering, Szechenyi Istvan University, 9026 Gyor, Hungary
\* Correspondence: mykola.sysyn@tu-dresden.de (M.S.); fischersz@sze.hu (S.F.)

**Abstract:** The share of rail transport in world transport continues to rise. As the number of trains increases, so does the load on the railway. The rails are in direct contact with the loaded wheels. Therefore, it is more easily damaged. In recent years, domestic and foreign scholars have conducted in-depth research on railway damage detection. As the weakest part of the track system, switches are more prone to damage. Assessing and predicting rail surface damage can improve the safety of rail operations and allow for proper planning and maintenance to reduce capital expenditure and increase operational efficiency. Under the premise that functional safety is paramount, predicting the service life of rails, especially turnouts, can significantly reduce costs and ensure the safety of railway transportation. This paper understands the evolution of contact fatigue on crossing noses through long-term observation and sampling of crossing noses in turnouts. The authors get images from new to damaged. After image preprocessing, MPI (Magnetic Particle Imaging) is divided into blocks containing local crack information. The obtained local texture information is used for regression prediction using machine-supervised learning and LSTM network (Long Short-Term Memory) methods. Finally, a technique capable of thoroughly evaluating the wear process of crossing noses is proposed.

**Keywords:** contact fatigue; neural convolution; machine learning; turnout; switch; frog rail; rail surface; long short-term memory

## 1. Introduction

Rail transportation plays a more critical role in the world transportation network. In this context, European railway traffic is multiplying, and railways keep Europe going. Railway operation safety has attracted more attention as a critical factor in ensuring smooth traffic. A European survey in 2018 showed that European local and regional passenger trains experience about one in ten travel delays. Many passenger and freight trains did not travel due to rail failure [1]. It will see that excellent and timely railway maintenance is of great significance to railway operation. The crossing is the main structure of the railway track to realize the transfer of trains, and it is also the weak link of the railway track and the critical facility for limiting the passing speed of the train [2]. The wheel-rail wear in the crossing is severe, and the service life is short, increasing the railway operation cost and posing a potential safety hazard. With the increase in train speed, the crossing's action becomes more complicated, which puts higher requirements on the quality of the crossing [3].

Most rail failures come from various damages on the rail surface, the most frequent failure. If someone can evaluate the rails' service conditions and the rails' status to predict the occurrence of rail failures, it will save a lot of inspection and maintenance costs. Timely

pre-grinding of the rail can also remove surface fatigue cracks and increase the service life of the rail [4]. As the weakest part of the rail system, it is imperative to detect and evaluate its surface state and make accurate predictions. Many scholars have studied the state of the ballast or subgrade to predict it, as shown in articles [5–8]. They are of reference significance to the state and evaluation of subgrade bearing capacity. Many scholars have carried out comprehensive research on various types of damage detection on the rail surface. However, there are few studies on the surface fatigue state of rails and crossings, and the evaluation and prediction of the state of crossings are rarer [9–11]. Moussa Hamadache et al. summarized the existing turnout detection methods, and machine vision detection is more environmentally friendly and efficient [12]. In Article [13] an evaluation of crossing modelling approaches has been performed in this article to justify their selection for the research interests of predicting the most dominant failure mechanisms of wear, rolling contact fatigue (RCF). For the surface fatigue detection and evaluation of crossings, the Mykola Sysyn team of the Dresden University of Technology proposed an effective research method in the article [14,15]. They obtain the crack image of the defects detection on the crossing nose surface. With the statistical regression algorithm, they successfully predict the location of the damage in advance 10 Mt (million tons) of the traffic volume and determine the state of the crossing nose. Vadim Korolev solve the issue of safe permissible dimensions of a gauge track and flangeway gap [16]. His team also reduce maintenance expenditures on curved turnouts and increase the travel speed through turnouts up to the ones when rounding a curve [17]. Based on the article [13], this paper conducts a further study. First, through the observation and sampling of the fatigue state of the surface of the crossing noses of multiple crossings, magnetic particle inspection is carried out. Finally, it can obtain the complete damage process of the crossing nose. Through the data of the damage process, information on defect states can be received. In order to avoid the error of crack feature extraction caused by natural conditions and other factors, this paper avoids the computationally crack extraction process. Instead, the authors extract texture features from the crossing nose image. The images need preprocessing. Firstly, the surface image of the crossing nose was divided into small images that can contain most of the surface cracks, and then 54 categories, including 33,696 texture features, were obtained through typical texture feature extraction algorithms for the images from 624 regionals. Finally, through various machine learning regression algorithms, as well as the CNN regression algorithm and LSTM regression algorithm, these eigenvalues are fitted and regressed, and the results to select the optimal regression algorithm can be compared. The calculation results can quite accurately distinguish the fatigue damage state of the crossing nose surface in four stages. The convergence index of the regression is optimal, which can accurately evaluate the fatigue state of the crossing nose. It also has a reasonable prospect of expanding application in other parts of the rail system. It provides a reliable research direction for damage prediction of the rail system.

## 2. Features Acquisition and Methods

After years of development, several mature enough rapid detection methods, such as electromagnetic induction and ultrasonic and magnetic particle inspection [18]. Traditional defect detection methods need to obtain specific information about damage, such as the length of cracks, the range of squat(s), and the size of spalling. However, after preprocessing the magnetic particle inspection images, this paper omits the complex crack detection process and only uses computer algorithms to obtain local texture features for analysis. As a result, it saves data processing time and has better prediction results than the literature [15].

### 2.1. A. Image Preprocessing

Magnetic particle inspection uses the interaction between the leakage magnetic field at the workpiece defect and the magnetic powder. It uses the difference between the magnetic permeability of the surface and near-surface defects (such as cracks, slags, hairlines, etc.) of steel products and the permeability of steel. After magnetization, the magnetic field at the

discontinuity of the material will be distorted. It will generate a leakage magnetic field on the surface of the workpiece, where part of the magnetic flux leaks. Thereby attracting the magnetic powder to form the magnetic powder accumulation on the defect magnetic trace, which shows the position and shape of the defect under appropriate lighting conditions, to observe and interpret the accumulation of these magnetic particles and realize magnetic particle inspection. The time and location of severe cracks and breakages in rails (especially in the crossing area) are random. In this study, high-definition image shooting and magnetic particle inspection at fixed traffic intervals were performed on multiple crossing areas until the crossing's complete rolling fatigue damage process was observed. The magnetic particle inspection image is obtained from the surface inspection of the crossing nose track, as shown in Figure 1.

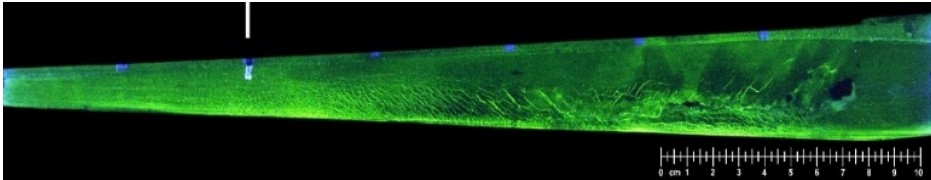

**Figure 1.** Image with magnetic particle inspection of a frog rail.

The original data in this paper is the complete and continuous destruction process of a small number of crossing noses obtained by Deutsche Bahn installing 11 crossings on a heavy traffic railway in Hannover after three years of uninterrupted observation. At the same time, the research team of China Railway Hohhot Group also provided data as an extension. The data of five moments in the whole service life cycle of the crossing from brand new to complete failure was recorded. After three basic image processing steps of image grayscale, noise reduction, and enhancement, each crossing nose can provide a set of MPIs data for the entire life cycle. The authors also took camera images of the rail surface during the same period for reference and verification. Figure 2 shows part of an example crossing nose data message.

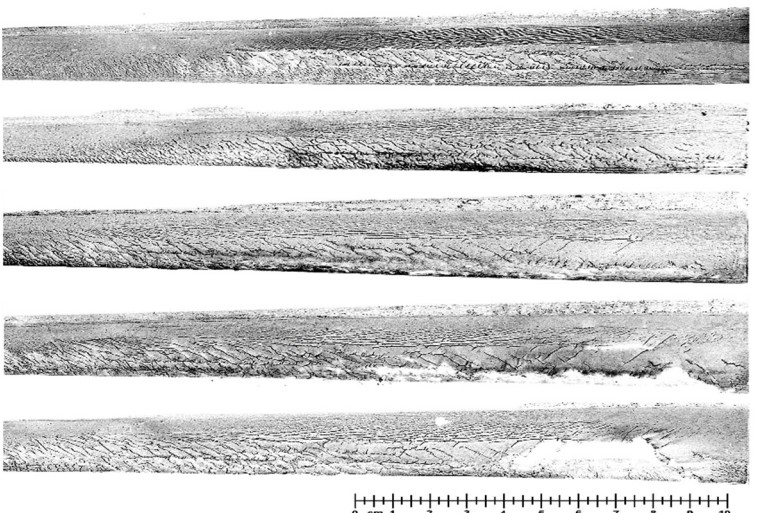

**Figure 2.** Sample dataset of a crossing nose.

The resulting image was unified to 1792 × 384 pixels and divided the image into 256 × 128 local patches in the 64-pixels step. The size of the partial image is selected according to the principle that the image of this size can completely contain most of the surface cracks of the crossing nose track. The segmented image obtained by the above method retains all the texture features. With the code of images, the specific position of the crossing nose track surface where the crack is located can be located, which provides

convenience for the subsequent return of results. Figure 3 shows the partially segmented image. Because of the geometry of the crossing nose track's widening cross-section with distance from the vertex, some segmented images contain more blank parts. For the accuracy of the data, this part of the local image was removed.

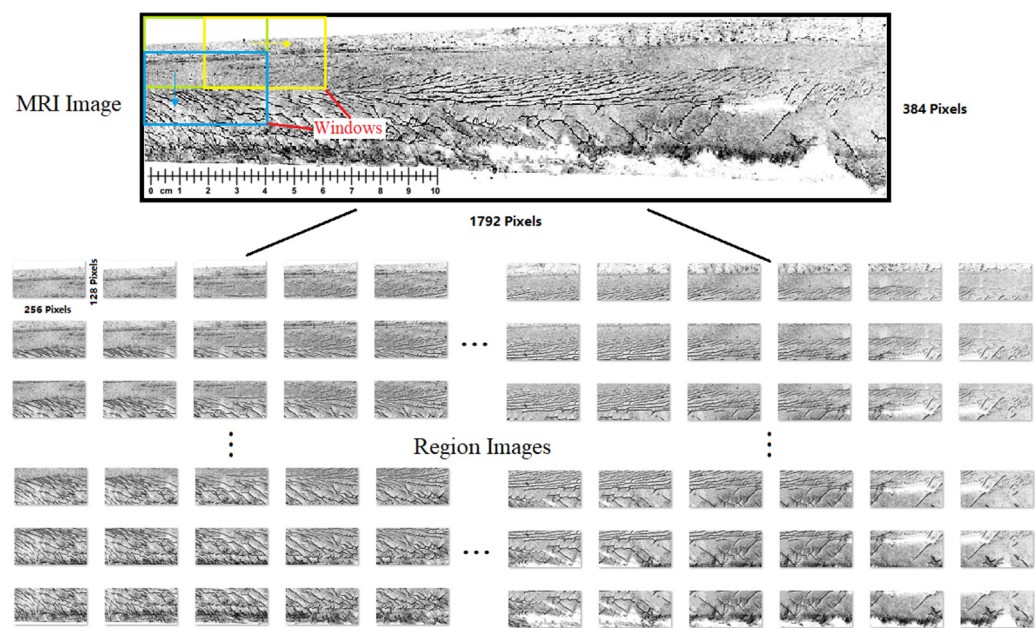

**Figure 3.** Region image acquisition.

### 2.2. Feature Extraction

The texture of an image is the quantized image feature in image computation. The image texture describes the spatial color distribution and light intensity distribution of an image or a region area within it. The authors usually refer to the repeated arrangement of some basic patterns as visual textures in image processing. Determining the tone units constituting the texture and determining the relationship between the tone units are the primary methods to describe texture information. The texture is a feature that exists in a local area. It is related to the size and shape of the area. Determining the boundary between two texture modes requires observing whether the value of the texture feature changes significantly. Texture reflects the structural framework of the information contained in the picture. The texture is an essential means of image segmentation, feature extraction, classification, and recognition. Statistical and structural methods are the primary means of texture analysis for spatial or transform domain images. Methods based on computer vision, such as Local binary patterns [19], Scale-invariant feature transforms [20], Haar-like [21], many image feature extraction.

However, feature extraction is mainly about finding feature points for visual detection and classification. These methods are not the same as the research purpose of this paper because the fractures change randomly and cannot correspond to one-to-one. General feature extraction mainly extracts the length, thickness, and number of cracks. The above method can be used when the crack structure is relatively simple and the number is small. Despite, in the face of a large amount of crack information and complex crack structures, the detection method of prominent texture features is more suitable. Because the extraction of texture features does not require specific crack objects, it is more in line with statistical results. More importantly, the detection method of texture features is faster than extracting detailed physical information and is not easily disturbed by external factors. Therefore, three detection methods of texture feature (GLCM, HOG, and Gabor) were chosen to be suitable for this paper and extract 54 feature values for each local image. All three methods have broad applications and validation.

### 2.2.1. GLCM Feature Extraction

In 1973, Haralick [22] proposed to describe texture features with a gray-level co-occurrence matrix (GLCM). The GLCM is a crucial method for analyzing image texture features, which has strong robustness and stability and is widely used in practical applications. The grayscale histogram results from statistics on a single pixel on the image having a certain grayscale. The grayscale co-occurrence matrix is obtained by statistics on the status of two pixels that maintain a certain distance on the image, respectively, having a certain grayscale. The joint probability density of pixels at two positions is the co-occurrence matrix. The distribution characteristics of brightness and the position distribution characteristics between pixels with the same or close brightness are the primary information in GCLM. GLCM is a second-order statistical feature matrix, the basis for defining a set of texture features. Its magnitude relates to changes in image brightness.

The gray level co-occurrence matrix of an image can reflect the comprehensive information of the gray level of the image on the direction, adjacent interval, and variation range. It is the basis for analyzing the local patterns of the image and their arrangement rules. The gray-level co-occurrence matrix starts from the pixel $(x, y)$ with the gray value of $i$ in the image to the pixel $(x + a, y + b)$ with the distance $d$, and the gray value is $j$. The authors count the $P(i, j, d, \theta)$. It is the frequency of simultaneous occurrence of $i$ and $j$.

$$P(i, j, d, \theta) = \{(x, y), (x + a, y + b) \mid f(x, y) = i, f(x + a, y + b) = j\} \tag{1}$$

It represents the grayscale co-occurrence matrix, an N $*$ N matrix (N is the gray level, which is the number of different grayscales or colors in a picture). Among them, $\theta$ is the generation direction of the gray level co-occurrence matrix, which usually goes in four directions of $0°$, $45°$, $90°$, and $135°$. In the GLCM matrix, if the elements near the diagonal have larger values than far elements, it means that the pixels of the image have similar pixel values. The gray level has a significant change locally. In order to describe the texture status with the co-occurrence matrix more intuitively, some parameters reflecting the matrix status from the co-occurrence matrix were derived. Typically as follows:

- Energy: It is the square sum of the element values of the gray co-occurrence matrix, so it is also called energy, which reflects the uniformity of the image's gray distribution and the texture's thickness.
- Contrast: it reflects the clarity of the image and depth of texture grooves.
- Correlation: It measures the similarity of the spatial grayscale co-occurrence matrix elements in the row or column direction. Therefore, the correlation value reflects the local grayscale correlation in the image.
- Entropy: It measures the amount of information an image has, and texture information belongs to the information of the image, which is a measure of randomness.
- Inverse differential moment: It reflects the homogeneity of the image texture and measures the local variation of the image texture. A significant value indicates no variation between different image text areas, and the local area is uniform.

This paper calculates the image matrix's mean, variance, and standard deviation of the above five main parameters. As a result, 15 features as reference data for regression analysis can be obtained.

### 2.2.2. HOG Feature Extraction

Histogram of Oriented Gradient (HOG) is a computational method that can be used to characterize the images. In computer vision detection, the detection target is locked by image processing [23]. Computing the gradient direction histogram of the local area of the image is a method for HOG to construct image features. The method of HOG feature analysis is typical in the field of image classification. HOG obtains the gradient feature of the image by calculating the size and direction of the gradient of each pixel in the image and is a feature descriptor. Due to the computation of local histograms and normalization,

it is remarkably invariant to image geometric and optical deformations. Figure 4 shows the processing process.

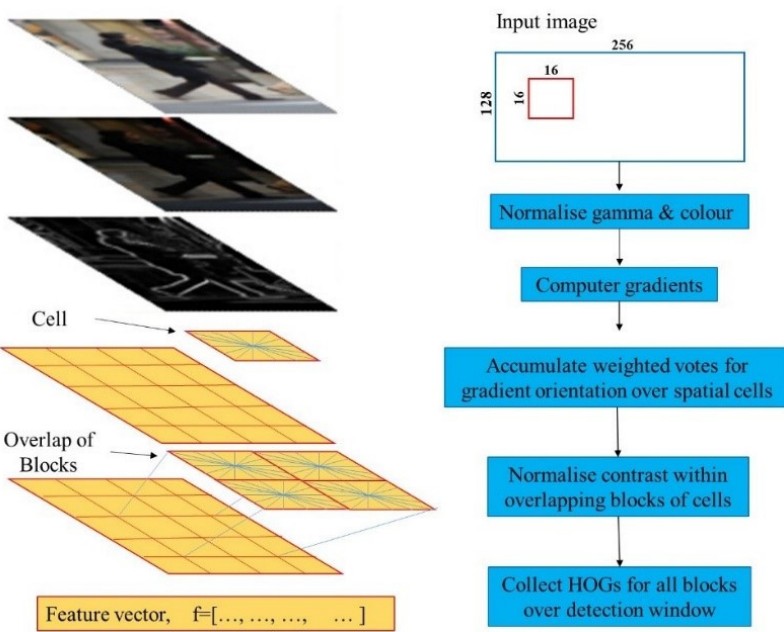

**Figure 4.** Process of HOG feature extraction.

The authors need to divide the image into several cells. In this paper, 16 × 16 pixels are one cell, and adjacent cells do not overlap. Unlike the traditional method, the magnitude of the gradient direction in each cell needed to be counted. All gradient directions were divided into 9-dimensional feature vectors. This method counts the cumulative gradient values corresponding to each angle. Each single cell unit group has nine values, connecting cells into a block. This way, the authors count the accumulated gradient values corresponding to each direction in a block. These values are the HOG features of this paper. However, there is an overlap between these blocks. All information is overlapped an equal number of times and does not affect the final statistical proportions. The final value has to be normalized. The value is the HOG description feature. Each image can get nine texture features. However, the crack feature studied in this paper is comprehensive data rather than a feature point, so the calculation method of the HOG feature is slightly modified here. In the calculation process of this paper, each block is a cell, so there is no overlapping area of data. The focus of the data is on the histogram channel values in nine gradient directions. Finally, the mean and variance of each matrix will be calculated separately. The final HOG features that make up the mobile phone are in this paper. In this way, the authors obtain 18 eigenvalues for each image corresponding.

### 2.2.3. Gabor Feature Extraction

Gabor filter has excellent advantages when extracting texture features of objects. It is very similar to the working principle of the optic nerve of the human visual system [24]. Gabor filters are good at extracting local spatial and frequency domain information from images. The Gabor wavelets have good sensitivity to the edges of objects. It can provide good orientation features and scale features. The insensitivity to illumination changes has little effect on the Gabor filter. These characteristics make the Gabor wavelet widely used in visual information analysis. It is a common means of computing technology sense detection. The two-dimensional Gabor wavelet transform is an essential tool for signal analysis and processing in the time-frequency domain. Its transform coefficients have good visual characteristics and biological backgrounds, so it is widely used in image processing, pattern recognition, and other fields. The complex sine function modulated by

the Gaussian function constitutes the Gabor function, which can extract a given region's local frequency domain features. The two-dimensional Gabor kernel function consists of a Gaussian function Multiplied by a cosine function, where $\theta$, $\phi$, $\gamma$, $\lambda$, $\sigma$ are parameters.

$$g_{\lambda,\theta,\varphi,\sigma,\gamma}(x,y) = \exp\left(-\frac{x'^2 + \gamma^2 y'^2}{2\sigma^2}\right)\cos\left(2\pi\frac{x'}{\lambda} + \varphi\right) \tag{2}$$

$$x' = x\cos\theta + y\sin\theta \tag{3}$$

$$y' = -x\sin\theta + y\cos\theta \tag{4}$$

Wavelength ($\lambda$): represents the wavelength parameter of the cosine function in the Gabor kernel function. Its value is specified in pixels, usually greater than or equal to 2, but not greater than 1/5 of the input image size.

Orientation ($\theta$): Indicates the orientation of the parallel strips in the Gabor filter kernel. Valid values are real numbers from 0° to 360°.

Phase offset ($\psi$): represents the phase parameter of the cosine function in the Gabor kernel function. Its value range is −180° to 180°. Among them, the equations corresponding to 0° and 180° are symmetrical with the origin, and the equations of −90° and 90° are centrosymmetric about the origin.

Aspect ratio ($\gamma$): The spatial aspect ratio, it determines the shape of the Gabor function. When $\gamma = 1$, the shape is circular; when $\gamma < 1$, the shape elongates with the direction of the parallel stripes. Usually, this value is 0.5.

Here $\sigma$ represents the standard deviation of the Gaussian factor of the Gabor function.

In this paper, the wavelength $\lambda$ is 16, and the direction $\theta$ are zero, $\pi/6$, $\pi/4$, $\pi/2$, $3\pi/4$, $5\pi/6$ and $\pi$. The relative phase offset $\psi$ is zero. The aspect ratio is 0.5. It can obtain seven sets of feature matrices in different directions. These data's mean, variance, and standard deviation were calculated again. Finally, the authors can get 21 texture features. In this paper, three texture extraction methods and 54 features are the primary data for analyzing and predicting the surface damage of frog tracks.

## 3. Regression Analysis

Although the direct extraction of texture information has saved much time compared to the direct extraction of crack and damage features, more than 30,000 data with 54 sets cannot be directly calculated and analyzed. With such massive data, it is difficult to ensure accuracy and efficiency using basic statistical analysis methods. Computer science has dramatically improved computing power and data storage capacity [25]. The advent of machine learning has provided more options for solving these types of problems. Machine learning has many types of tasks, but supervised, and unsupervised learning can generally be used. The focus of the supervised learning model of machine learning is to predict the target/marker of an unknown sample based on the knowledge of existing experience. Unsupervised learning tends to analyze the characteristics of things themselves, and commonly used techniques include Dimensionality Reduction and Clustering. This paper explores the observation and prediction of rolling fatigue damage on the rail surface, which is the analysis and prediction of continuous variables. Therefore, the supervised regression prediction algorithm is suitable for the research data in this paper. The deep learning algorithm of the neural network also has this wide application and a good effect on regression prediction [26,27]. Therefore, the authors use these two methods to regress the data and compare the results.

### 3.1. Machine Learning Regression

The most common examples of supervised machine learning are Support Vector Regression and Tree Ensemble. Both algorithms are prevalent and mature. Gaussian process regression (GPR) is generally regarded as a highly expressive supervised learning algorithm comparable to the deep neural network (DNN). GPR has the excellent intuitive property that all interpolated mean predictions are generated as a weighted linear combination of

the existing mean points in the training set, as measured by the distance from the test point to a given data point (measured in the space of the kernel function) zoom. [28]. GPR require less data because they have fewer parameters to tune. However, with more data, significantly when the density does not increase over the fixed domain [29], which can help significantly improve performance. Predicted values are probabilistic (Gaussian), so the empirical confidence intervals can be calculated and then, based on this information, re-fit (online fitting, adaptive fitting) predictions in a particular region of interest. Moreover, GPR is robust to phenomena such as exploding and vanishing gradients. Different cores can be specified. Standard kernels are provided, but specific kernels can also be selected.

Gaussian process regression (GPR) models are nonparametric kernel-based probabilistic models with a finite collection of random variables with a multivariate distribution [30]. The Gaussian process has one more time or space dimension than the Gaussian distribution or one more prediction process. Figure 5 shows the steps of using Gaussian process regression prediction.

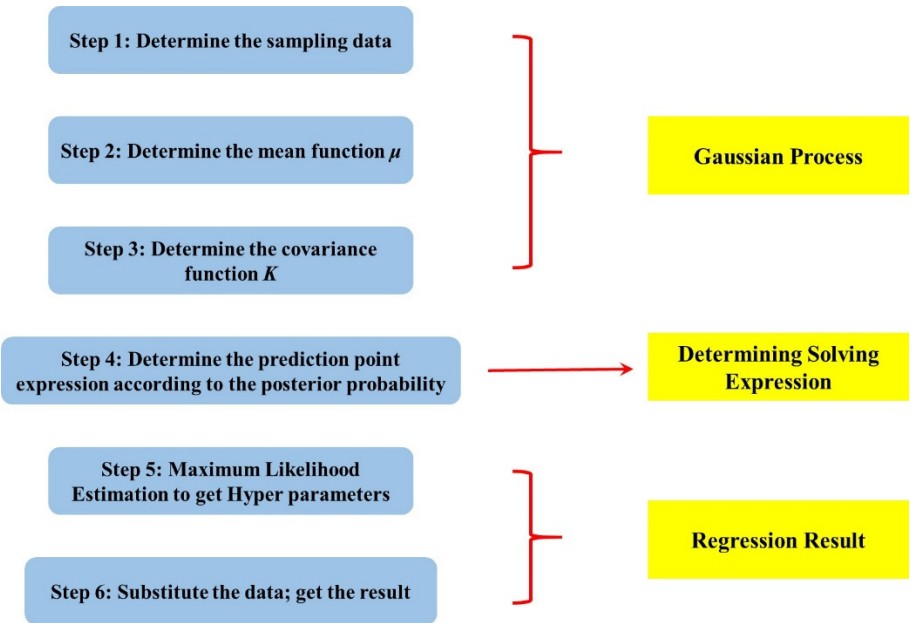

**Figure 5.** Gaussian Process Regression.

This paper uses three algorithms for regression processing of the same dataset in SVR, Tree Ensemble, and GPR. First, the results of the above three methods are compared, and the best-supervised machine learning algorithm is selected. Then the authors compare the machine-learning algorithm with the results of deep learning to ensure that the final result obtained in this paper is optimal.

*3.2. Deep Learning Regression*

Convolutional Neural Networks (CNN) are best at processing images. The human optic nervous system inspires it. Currently, the application of CNN is the most extensive in deep learning, such as face recognition, automatic driving, security, and many other fields. However, what has sprung up for regression problems is Recurrent Neural Networks (RNN). RNN is a class of artificial neural networks that have become more popular recently. RNN learns from ordered data, and RNN will remember the last data like a human but sometimes forget the previous data [31]. To solve this drawback of RNN, In 1997, Sepp Hochreiter and Jürgen Schmidhuber presented LSTM networks [32], whose full English name is Long short-term memory, which is also one of the most popular RNNs. Compared with the simple RNN structure, LSTM is an excellent variant model of RNN. It inherits the characteristics of most of the RNN models and solves the Vanishing Gradient problem

caused by the gradual reduction of the gradient backpropagation process. In addition, LSTM introduces a "gate" structure, which effectively avoids terrible conditions such as gradient explosion and gradient dispersion, and thus has better prediction and fitting performance for data. LSTM is popular in image recognition, semantic understanding, time series analysis, and other fields. In comparisons with other nets, Elman nets, and Neural Sequence Chunking, LSTM leads to many more successful runs, and learns much faster. LSTM also solves complex, articial long time lag tasks that have never been solved by previous recurrent network algorithms. Structures called "gates" are carefully designed parts of an LSTM. It can remove or add information to the cell state. The primary function of the gate is to select suitable information. It consists of a pointwise multiplication operation and a sigmoid neural network layer. A single-layer module repeat calculates in standard RNN. LSTM-Network has a similar structure to RNN.

Nevertheless, the repeated modules of the LSTM have a unique structure, as shown in Figure 6. It comprises four neural structures that communicate information in particular ways. Figure 6 illustrates the architecture of a simple LSTM network for regression. The network starts with the sequence input layer and then the LSTM layer. Finally, a fully connected layer and a regression output layer are at the network's end.

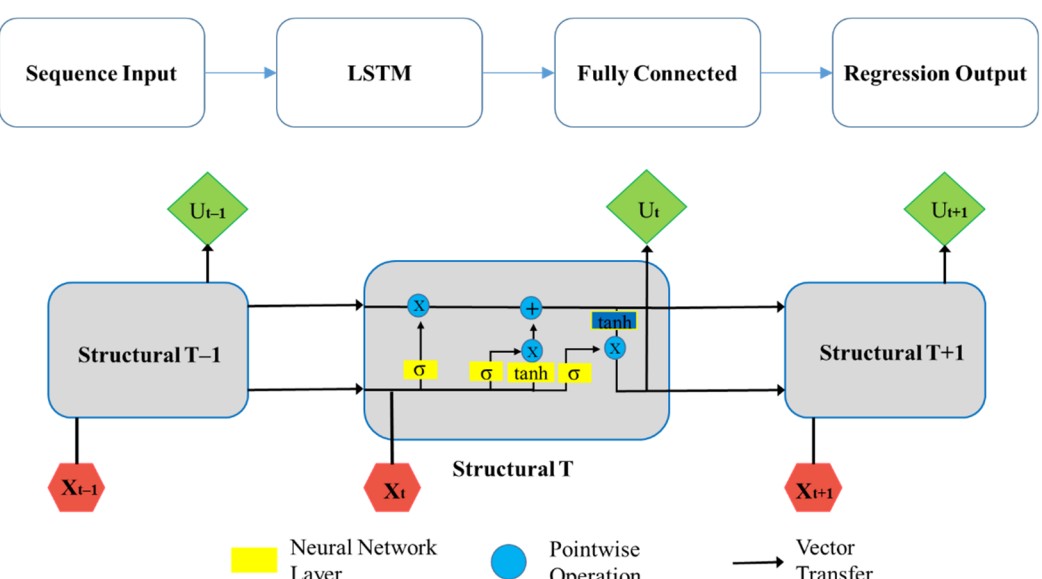

**Figure 6.** Structure of the LSTM network.

The specific steps of LSTM are too complicated. This paper makes a simple analysis in Figure 7. When the input is input to the hidden layer of the LSTM network, it is firstly transformed by the input gate and then superimposed with the memory cell state processed by the forget gate to form a new memory cell state. Finally, the memory cell state is processed by the nonlinear function. The output of the hidden layer can be obtained by dot product with the current information state processed by the nonlinear function. At the same time, the ordinary CNN model can also realize the regression algorithm of the data. Therefore, it can be called an RNN network. In order to verify that the calculation results of LSTM in predicting the evolution of fatigue on crossing nose surfaces are relatively superior, this paper will also build a CNN model for regression prediction. This paper compares the results of the two with those calculated by machine-supervised learning, allowing us to find a relatively perfect fit regression model.

**LSTM Step-by-Step Process**

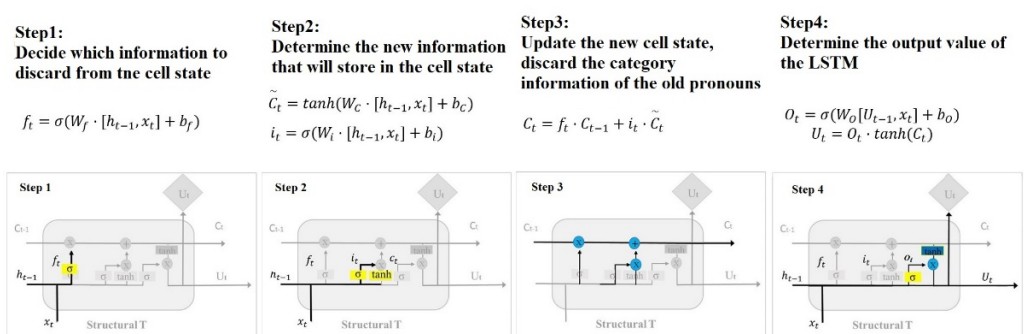

**Figure 7.** LSTM step-by-step process.

*3.3. Evaluation Indicators*

In regression tasks (prediction of continuous values), common evaluation metrics are Root Mean Square Error (RMSE) [33], Mean Squared Mean Square Error (MSE) [34], and Mean Absolute Error (MAE) [35]. RMSE has the same dimension as MAE, but after calculating the result, the authors will find that RMSE is larger than MAE. RMSE is to square the error first and then extract a root, which magnifies the gap between larger errors. The MAE reflects the true error. Therefore, in the measurement, the smaller the value of RMSE, the better the fitting because its value can reflect its maximum error and is relatively small. The problem with the above measurement methods is that there is no upper and lower limit, and the coefficient of determination) $R^2$ [36] solves this problem. This article does not discuss their differences in detail and directly uses these data to measure the effect of regression models.

The smaller the first three values are, the better the fitting effect. The larger the $R^2$ is, the higher the degree of explanation of the independent variable to the dependent variable. At the same time, the higher the percentage of the change caused by the independent variable to the total change. The scatter points are more clustered near the regression line (note that $R^2$ is not necessarily a 1:1 line), and in general, when $R^2$ is higher than 0.8, the fitting effect is better.

## 4. Results

Observing many variables that reflect things is a problem that modern scientific research often faces. Scholars must collect large amounts of data for analysis and then look for patterns. A large number of multivariate samples undoubtedly provide rich information for research and application, but it also increases the workload of data collection to a certain extent. Usually, many associations exist between variables, which increases the difficulty of snooping on patterns and brings inconvenience to analysis. Principal component analysis (PCA), is mainly used for dimensionality reduction operations on high-dimensional data to extract the main features of the data and remove useless or noisy information. In this paper, the process of reducing the dimension of the experimental data through PCA is expected to obtain a reasonable feature dimension. The first six main features in the PCA-calculated data were selected and gradually increased to 54. The four evaluation indicators of the experimental results are then compared to determine the optimal feature selection result.

Table 1 is mainly to confirm the number of features that need to be selected in this paper. After sorting by feature influence, increase the number of features in order to observe the changes of the corresponding four parameters, so as to select the optimal number of features. It is observed from the results in Table 1 that the values of RSME, MSE, and MEA decrease significantly at the beginning as the number of selected features increases. Then it starts to slow down in the range of the number of features from 27 to 44. When the number of features increases to 54, there is a significant decline again and to the minimum. In contrast, $R^2$ has the opposite trend with these three values. They are reducing the number

of features to reduce the calculation time. Eighteen principal components can be chosen for analysis when the data is vast because the $R^2$ value has exceeded 80%. However, to get the best results, this paper selects all 54 features for machine learning.

**Table 1.** Variation of the Indicators as the Number of Features Increases.

| Feature No. | 6 | 12 | 18 | 27 | 35 | 44 | 48 | 52 | 54 |
|---|---|---|---|---|---|---|---|---|---|
| RSME | 7.314 | 6.2298 | 4.4361 | 4.1066 | 4.1909 | 3.9339 | 3.8559 | 3.8628 | 3.3273 |
| MSE | 53.495 | 38.811 | 19.679 | 16.864 | 17.563 | 15.475 | 14.868 | 14.921 | 11.071 |
| MEA | 5.2493 | 4.3527 | 3.1284 | 2.9974 | 3.0312 | 2.8421 | 2.8344 | 2.8266 | 2.3086 |
| $R^2$ | 0.58 | 0.7 | 0.85 | 0.87 | 0.86 | 0.88 | 0.88 | 0.88 | 0.91 |

*4.1. Machine Learning Results*

SVR and Tree Ensemble have supervised learning methods for regression. SVR uses the principle of Statistical Risk minimization to fit the regression line. Tree Ensemble cuts the data set into many easily modeled data and then uses linear regression techniques for modeling. Similar to GPR, these two algorithms also need a kernel function. Algorithms lacking contrast are not enough to justify the choice. In this paper, the above two algorithms and the GPR algorithm are used to perform regression analysis on the surface texture data of the crossing nose. The authors incorporate multiple kernel functions into each algorithm. Finally, the regression prediction results of machine learning in 11 are obtained. This paper lists the four evaluation indicators of the 11 models in Table 2 for comparison. The optimal machine-learning model was selected. The purpose of Table 2 is to select the optimal unsupervised learning algorithm by comparison. This algorithm is then compared with the algorithm of the deep neural network. The optimal algorithm may be selected after multiple comparisons. From the results in Table 2, it can be seen that the regression model of the GPR algorithm has advantages. The GPR algorithm, whose kernel function is Rational Quadratic, has the best results compared with other machine learning algorithms. The values of RSME, MSE, and MEA reach the minimum, while $R^2$ reaches the maximum, up to 0.91. This result can be fully applied to actual detection and prediction. The results in Tables 1 and 2 also show that RSME is almost proportional to the other two indicators (MSE and MEA). Therefore, the authors only need to refer to one indicator. $R^2$ and RSME were applied as comparison metrics for neural network results.

**Table 2.** Machine Learning Regression Comparisons.

| | GPR | | | | Tree Ensemble | |
|---|---|---|---|---|---|---|
| | Squared Exponential | Matern 5/2 | Rational Quadratic | Exponential | Bagged Trees | Boosted Trees |
| RSME | 3.5518 | 3.3739 | 3.3273 | 3.9627 | 4.9450 | 5.0920 |
| MSE | 12.6160 | 11.3830 | 11.071 | 15.7370 | 24.4530 | 25.9290 |
| MEA | 2.5092 | 2.3460 | 2.3086 | 2.8581 | 3.5040 | 3.8013 |
| $R^2$ | 0.90 | 0.91 | 0.91 | 0.88 | 0.81 | 0.80 |
| | | | SVR | | | |
| | Linear | Quadratic | Cubic | Fine Gaussian | Medium Gaussian | Coarse Gaussian |
| RSME | 7.0201 | 28.528 | 21.025 | 8.7155 | 5.4601 | 8.3760 |
| MSE | 49.282 | 62.847 | 42.199 | 73.787 | 27.408 | 68.912 |
| MEA | 5.1181 | 3.6229 | 3.6891 | 7.2957 | 3.8202 | 6.0487 |
| $R^2$ | 0.62 | 0.51 | 0.67 | 0.43 | 0.79 | 0.46 |

### 4.2. Long Short-Term Memory Results

The input layer of the LSTM network in this chapter is 54 texture feature data, and the LSTM layer contains a total of 200 hidden units. At the end of the network is a fully connected layer. Finally, the regression result is output through the regression layer. During this period, the data needs to go through 250 rounds of training. Specify an initial learning rate of 0.005 and reduce the learning rate after 125 epochs by multiplying by a factor of 0.2. To prevent exploding gradients, the authors set the gradient threshold to 1. Figure 8 shows the training process of LSTM.

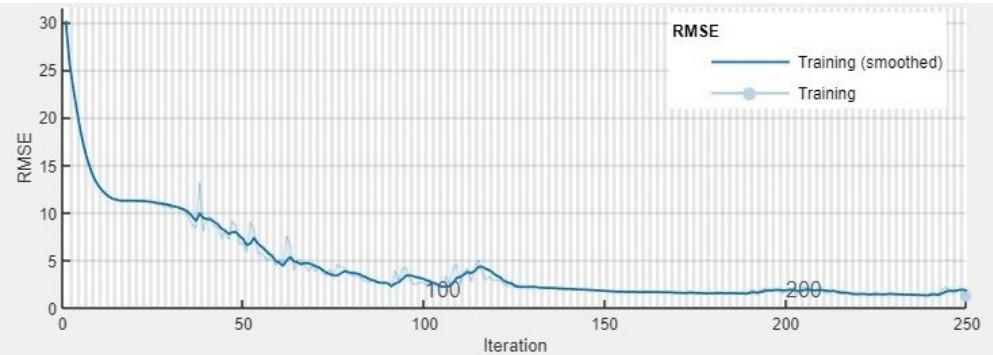

**Figure 8.** The training process of LSTM.

The RMSE value of LSTM keeps decreasing until it reaches a meager value. In contrast, CNN implements regression prediction. The input data is in the form of m × n. here m represents how many input features there are (54 features). n is how many samples (624 data samples). The output data is "one × n", and 1 (one) represents the output is a single output. After the convolutional layer and the activation function are two fully connected layers with 384 neurons. The final regression layer outputs the regression results. As the optimal solution in supervised learning, GPR is compared with the regression model of CNN and LSTM. Figure 9 compares the scatter plots formed by the predicted values of the three models with the actual values.

The data in Figure 9 shows a damaging process on the surface of a crossing nose. The vertical axis is the traffic volume. The original data is the texture data of the crossing nose surface when the traffic volume is 13 Mt, 22 Mt, 33 Mt, and 43 Mt. The crossing nose surface was damaged entirely when the traffic volume was 52 Mt. Because the image of a wholly broken MPI will cause blank areas to cause missing crack information, only the texture data before the complete breakage is included in the analysis. In the results of the three regression prediction models in Figure 8, the predicted points of GPR and CNN have a specific error with the actual value, but the results are relatively accurate. The results of the two models form a cluster of predictions for each period, and there are individual points with significant errors. However, the predicted result of LSTM is relatively continuous, and the error of the actual value is small and within a reasonable range. The results in Figure 9 indicate that LSTM has the best regression fit.

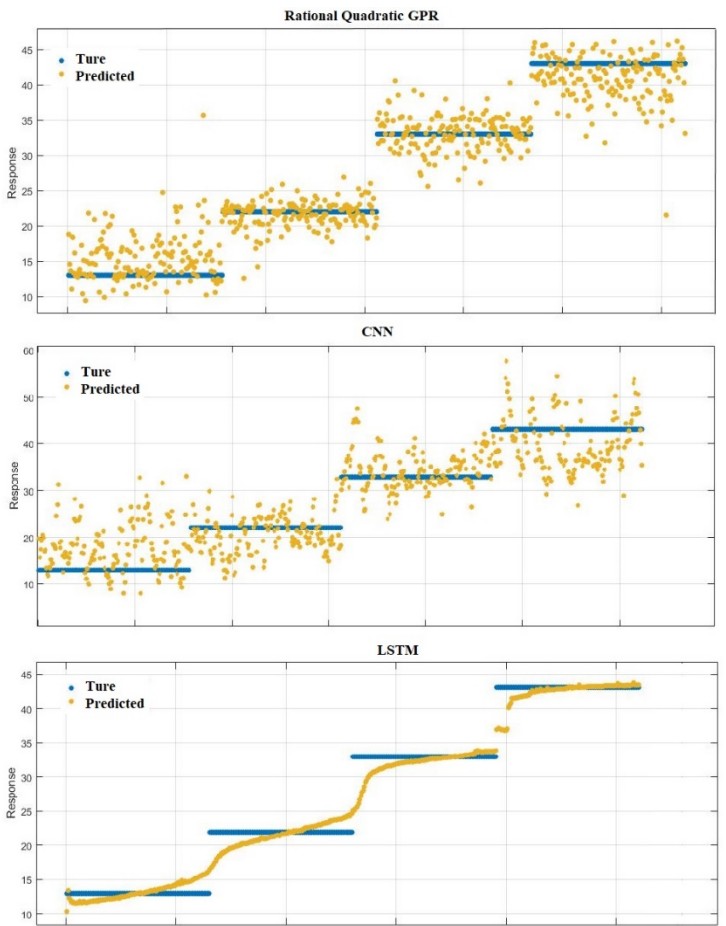

**Figure 9.** Comparison of regression models.

### 4.3. Results Evaluation

Figure 10 is the regression prediction result of LSTM. This result is not only smaller than the RMSE, the leading indicator of GPR but also achieves a high value of 0.9756 for $R^2$. The results of this paper have higher sensitivity than the results of [15]. The size of the main index proves that the result can evaluate and predict the damaged state of the track surface at any time. However, literature [15] can only expect in advance the surface damage of crossing nose tracks with a traffic volume of 10 Mt.

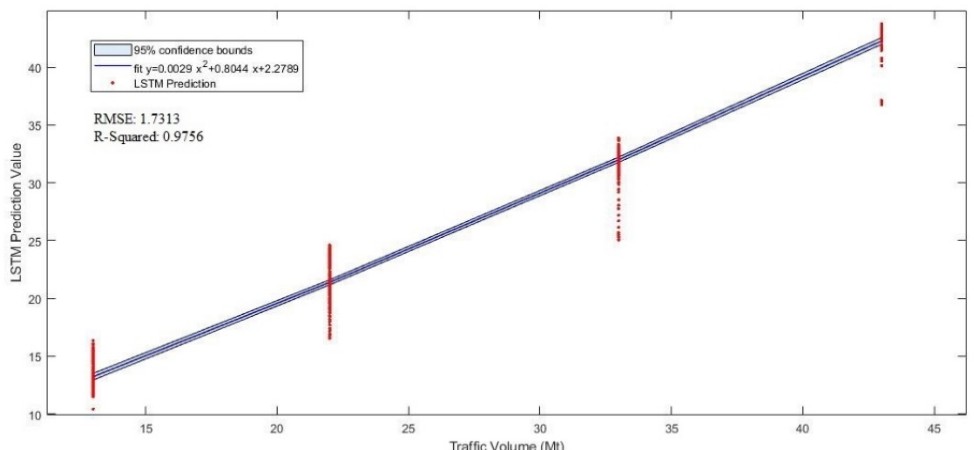

**Figure 10.** Regression metrics for LSTM.

The discrete points in Figure 9 are the red data points in Figure 10. The blue curve and light blue confidence intervals in Figure 10 are the regression predictions for the breaking time of crossing nose surface. The crack change is a continuous process corresponding to the constant evolution of the blue curve. Although the calculated result has a large dispersion, it is enough to measure the fatigue degree of the rail surface and judge whether the area needs to be polished in time.

Applying the LSTM's regression prediction results to evaluate the crossing nose's current fatigue state is not intuitive enough. In this paper, the calculated value of the regression function is converted to a more intuitive expression of the gradient color. As shown in Figure 11, the rail surface image of each traffic volume is divided into 168 partial photos. After the regression function calculation of LSTM or CNN, each analysis result is a long rectangular image composed of a group of 168 small color blocks. Take the CNN regression result with a traffic volume of 43Mt as an example. Each small color block in the large rectangle corresponds to the predicted value of a region image in Figure 2. According to the arrangement position of the local images in Figure 2, the corresponding color blocks are mapped to the 43Mt regression results. Therefore, the eight rectangular bars in Figure 11 are the independent results obtained from the 168 partial images calculated by the regression function. There are a total of four measurement data and two analysis methods, which correspond to the verification results of the regression function on 1344 local images.

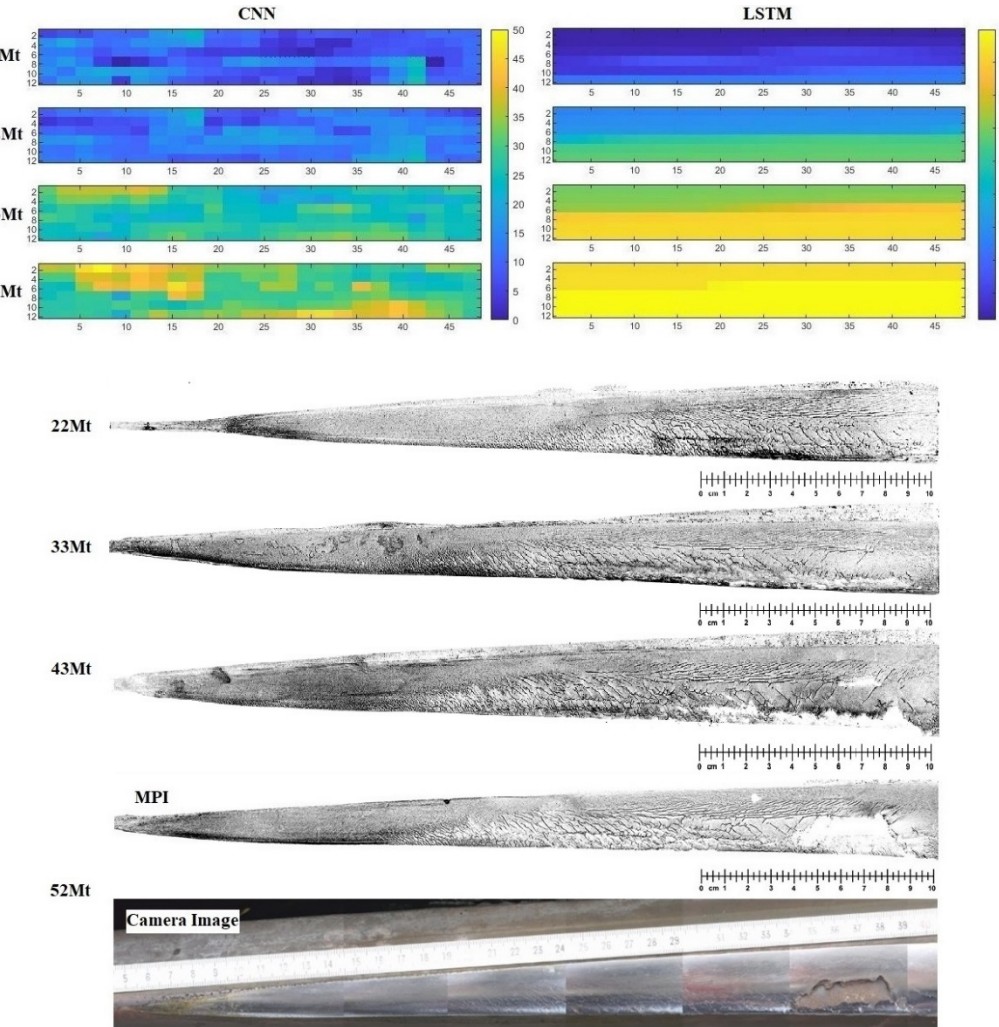

**Figure 11.** Predicting coloration of surface damage on crossing nose.

The results are sufficiently general and applicable. The more serious the surface damage of the crossing nose, the higher the predicted value and the brighter the color. Dark blue indicates no dangerous rolling fatigue cracks on the surface of the crossing nose. The color changes from blue to green to yellow as the traffic volume increases. As a sample crossing nose to display, it needs to be replaced at 43 Mt, and the bright yellow warning is in the picture. When it reaches 52 Mt, as shown at the bottom of Figure 11, the surface of the crossing nose has a severe block drop and ultimately fails. Figure 11 also compares the CNN model further with the LSTM. In Figure 9, the calculation results of the CNN and GPR models are similar. In Figure 11, the calculation result of LSTM corresponds to the color change. It is more continuous and smooth. The results of CNN and LSTM are similar, verifying the feasibility of predicting crossing nose failures through crack features. But the results of LSTM are more explicit and can better identify the surface state of the crossing nose at this time. We add the magnetic particle inspection pictures corresponding to the accumulated traffic volume in Figure 11 as a verification. Combined with the evaluation of DB experts, there are a large number of complex structural cracks and surface spalling in the 43 Mt magnetic particle inspection results, and the crossing nose is in an early warning state at this time. However, the LSTM algorithm in this paper passes a sufficient number of yellow blocks, that warn of failures at 33 Mt. Sure enough, at 52 Mt, there were large areas of falling blocks on the surface of the crossing nose, which seriously affected driving safety.

The regression prediction results of LSTM are more accurate and timely. At the same time, the side with more traffic on the same crossing nose surface is also brighter. In practical applications, when the predicted result of the fatigue defect is between green and yellow, a grinding process on the crossing nose surface was taken. The predicted value of the polished crossing nose turns blue predictably. Not only can the authors find the crossings that need grinding in time, but also the maintenance costs can be saved, and the energy consumption can be reduced. The crack change is continuous, but the measurements are spaced. Using the regression function to convert the measurement results into corresponding colors, when the blue color block is small, and the yellow color block becomes more, it indicates that the rail will be seriously damaged. At this time, grinding the crossing nose's surface can effectively avoid severe damage and prolong the service life. However, the predicted advance cannot be used as a reference. When the yellow patches appear, the fatigue has accumulated to a certain extent. When the green patches disappear, severe damage may soon occur. To ensure the railway operation safety, it is best to perform maintenance on the crossing nose before the green color patch disappears when applying this paper's results.

## 5. Conclusions

The research on the evolution of surface cracks in the whole service cycle of crossing noses in this paper shows that the characteristic changes of cracks can be obtained by extracting texture data. The problem extraction algorithms of GLCM, HOG and the Gabor filter are suitable for rolling fatigue research on the rail surface. The texture data on the surface of the crossing nose has a significant positive correlation with the life cycle of the crossing nose. In this paper, the fatigue damage information about crossing noses is obtained by only extracting the surface texture features of crossing noses. This method saves the time of calculation and processing and has a broader scope of application. Among machine learning algorithms, the regression prediction of GPR algorithm data has the best results.

On the other hand, the convolutional neural algorithm of LSTM is more suitable for predicting fatigue damage on the surface of crossing noses. The predicted results of the LSTM algorithm not only have stronger sensitivity than other algorithms and correspond continuously to the actual results. Therefore, the algorithm can realize a real-time prediction in practice, improving railway operation safety and reducing maintenance costs.

However, this method cannot be practical because acquiring MPI images takes a long time in practical applications. It is a significant disadvantage. The regular operation of railways has often been open to traffic, and if it takes too long to collect data, it will have a

severe impact as a future research question. The authors want to replace MPI images with camera images or videos. The LSTM model combined with CNN can directly extract image features through high-definition images, reducing computing time. However, the above methods require data over a long time and capital investment.

**Author Contributions:** Conceptualization, M.S. and S.F.; methodology, L.K., Y.H. and D.P.; software, J.L.; validation, J.L., O.N. and L.K.; investigation, L.K., Y.H. and D.P.; resources, M.S.; data curation, M.S.; writing original draft preparation, L.K.; writing—review and editing, S.F.; visualization, O.N., Y.H. and D.P.; supervision, M.S. and S.F. All authors have read and agreed to the published version of the manuscript.

**Funding:** This research received no external funding.

**Institutional Review Board Statement:** Not applicable.

**Informed Consent Statement:** Not applicable.

**Data Availability Statement:** The datasets used and/or analyzed during the current study are available from the corresponding author upon reasonable request.

**Conflicts of Interest:** The authors declare no conflict of interest.

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
