# Peer review of "Evolution of Rail Contact Fatigue on Crossing Nose Rail Based on Long Short-Term Memory"

_sustainability, doi:10.3390/su142416565_

Round 1

Reviewer 1 Report

This work uses different algorithms to deal with the MPI data to predict rail contact fatigue evolution. Generally, it is well-considered and of good reference value to scholars in this field.

There are a few issues that need to be improved before publishing.

1. Some cross citations are wrong. For instance, on page 13, line 456, figure 8 should be 9; on page 14, line 483, figure 10 should be 11.

2. Figure 9, the blue lines on the diagram are actually not realistic. A better method or assumption should be considered.

3. The results with CNN and LSTM (see Figure 11) lack explanation and argumentation. 

4. The results are few and lack validation as well as detailed evaluation, more results and analysis must be implemented!!

Author Response

Reviewer 1:

  1. Some cross citations are wrong. For instance, on page 13, line 456, figure 8 should be 9; on page 14, line 483, figure 10 should be 11.

Response: Thanks to the editor's careful review, we have corrected the errors as requested.

  1. Figure 9, the blue lines on the diagram are actually not realistic. A better method or assumption should be considered.

Response: Thanks for the editor's suggestion, but this plot is automatically generated by MATLAB and I cannot change it. It's not a blue line here, it's actually a blue dot. Corresponding to each discrete point, Yellow points are predicted values. There are true values at the same abscissa position indicated by blue dots.

  1. The results with CNN and LSTM (see Figure 11) lack explanation and argumentation.

Response: Thanks to the editor's suggestion. The added description is as follows:

The results of CNN and LSTM are similar, verifying the feasibility of predicting crossing nose failures through crack features. But the results of LSTM are more explicit and can better identify the surface state of the crossing nose at this time. We add the magnetic particle inspection pictures corresponding to the accumulated traffic volume in Fig. 11 as a verification. Combined with the evaluation of DB experts, there are a large number of complex structural cracks and surface spalling in the 43Mt magnetic particle inspection results, and the crossing nose is in an early warning state at this time. However, the LSTM algorithm in this paper passes a sufficient number of yellow blocks, that warn of failures at 33Mt. Sure enough, at 52Mt, there were large areas of falling blocks on the surface of the crossing nose, which seriously affected driving safety.

4.The results are few and lack validation as well as detailed evaluation, more results and analysis must be implemented!!

Response: Thanks to the editor's suggestion, We have added magnetic particle inspection pictures to compare and verify the results. Furthermore, each image is an independent verification result. The rectangular bars of CNN and LSTM in Fig. 11 are the independent results obtained from the 168 partial images calculated by the regression function. There are a total of four measurement data and two analysis methods, which correspond to the verification results of the regression function on 1344 local images.

We add the magnetic particle inspection pictures corresponding to the accumulated traffic volume in Fig. 11 as a verification. Combined with the evaluation of DB experts, there are a large number of complex structural cracks and surface spalling in the 43Mt magnetic particle inspection results, and the crossing nose is in an early warning state at this time. However, the LSTM algorithm in this paper passes a sufficient number of yellow blocks, that warn of failures at 33Mt. Sure enough, at 52Mt, there were large areas of falling blocks on the surface of the crossing nose, which seriously affected driving safety.

Reviewer 2 Report

(1) As the weakest part of the track system, switches are more prone to damage. The object of this paper is well chosen, and it is valuable for this research.  The title of this paper is not accurate and suitable to cover the content of this paper.

(2) In the section of Introduction, relevant references are not enough provided and stated. 

(3) It is not enough to analyze data in Table.1 and Table.2.

Author Response

Reviewer 2:

  • As the weakest part of the track system, switches are more prone to damage. The object of this paper is well chosen, and it is valuable for this research.  The title of this paper is not accurate and suitable to cover the content of this paper.

Response: Thanks to the editor's careful review, we have corrected the title of this paper.

  • In the section of Introduction, relevant references are not enough provided and stated.

Response: Thanks to the editor's suggestion, we added 4 references.

  • It is not enough to analyze data in Table.1 and Table.2.

Response: Thanks to the editor's suggestion. The added description is as follows:

Table 1 is mainly to confirm the number of features that need to be selected in this paper. After sorting by feature influence, increase the number of features in order to observe the changes of the corresponding four parameters, so as to select the optimal number of features.

The purpose of Table 2 is to select the optimal unsupervised learning algorithm by comparison. This algorithm is then compared with the algorithm of the deep neural network. The optimal algorithm may be selected after multiple comparisons. From the results in Table 2, it can be seen that the regression model of the GPR algorithm has advantages.

Reviewer 3 Report

This work understands the evolution of contact fatigue on crossing noses through long-term observation and sampling of crossing noses in turnouts. Also, a technique capable of thoroughly evaluating the wear process of crossing noses is proposed. In some cases, the text must be clarified and there are issues that are not accurate. Better and more evidences for some conclusions are required.

1.     Figure.2&3 should add scales.

2.     All the data should add error bars.

3.     How to verify accuracy rating of the prediction.

Author Response

Reviewer 3:

  1. Figure.2&3 should add scales.

Response: Thanks to the reviewer's suggestion, we added scales in Fig 2 and 3.

  1. All the data should add error bars.

Response: Thanks to the reviewer's suggestion, Figure 10 has error bars, but Figure 9 is automatically generated by MATLAB, and we cannot add it separately.

  1. How to verify accuracy rating of the prediction.

Response: Thanks to the reviewer's careful review, each image is an independent verification result. The rectangular bars of CNN and LSTM in Fig. 11 are the independent results obtained from the 168 partial images calculated by the regression function. There are a total of four measurement data and two analysis methods, which correspond to the verification results of the regression function on 1344 local images.

We add the magnetic particle inspection pictures corresponding to the accumulated traffic volume in Fig. 11 as a verification. Combined with the evaluation of DB experts, there are a large number of complex structural cracks and surface spalling in the 43Mt magnetic particle inspection results, and the crossing nose is in an early warning state at this time. However, the LSTM algorithm in this paper passes a sufficient number of yellow blocks, that warn of failures at 33Mt. Sure enough, at 52Mt, there were large areas of falling blocks on the surface of the crossing nose, which seriously affected driving safety.

Reviewer 4 Report

The paper reported the prediction of rail contact fatigue evolution on switch frog rail based on LSTM. The results and data are of great importance in engineering application. However, some technical errors exist. It can be accepted after minor revision.

Specific Comments:

1. The ruler should be given in Fig. 2 and Fig. 3, for a clear understanding.

2. The colors of two curves in Fig. 8 should be distinct.

3. In title, the LSTM should be revised into full name.

4. The machine-supervised learning and LSTM network (Long Short-Term Memory) methods should be further described in the introduction section. In other words, why was this method selected for prediction?

5. How is the fatigue damage characterized and determined in this work? This should be explained.

6. How were the experimental data compared with calculation results? We did not see obvious experimental data correlated with fatigue damage.

7. It is suggested that "this method can be extended to all surface damages of rails" should be erased in the abstract since the evidence is not sufficient.

Additional comments:

1)     The grammar of the sentence on lines 148-150 is wrong and it should be corrected. In addition, the word “etc.” should be deleted.

2)     On line 401, the incorrect writings should be corrected. For example, “80\%” should be “80%.”

3)     On line 407, “Of” should be “of.”

4)     The grammar of the sentence on line 442 is wrong since there are lack of conjunctions between individual sentences.

5)     Some irregular writings in the manuscript are required to be carefully examined and corrected. For example, on line 448, and line 456, the word “Figure” should be written in the form of “Fig.”

Author Response

Reviewer 4:

Specific Comments:

  1. The ruler should be given in Fig. 2 and Fig. 3, for a clear understanding.

Response: Thanks to the reviewer's suggestion, we added scales in Fig 2 and 3.

  1. The colors of two curves in Fig. 8 should be distinct.

Response: Thanks for the reviewer's suggestion, but this plot is automatically generated by MATLAB and I cannot change it.

  1. In title, the LSTM should be revised into full name.

Response: Thanks for the reviewer's suggestion, I have change it.

  1. The machine-supervised learning and LSTM network (Long Short-Term Memory) methods should be further described in the introduction section. In other words, why was this method selected for prediction?

Response: Thanks for the reviewer's question.

In comparison with other nets, LSTM leads to many more successful runs, and learns much faster. LSTM also solves complex, artificial long time lag tasks that have never been solved by previous recurrent network algorithms. We are not simply choosing LSTM. After comparing with 11 other unsupervised learning methods and the traditional CNN network structure, LSTM achieved the best results.

  1. How is the fatigue damage characterized and determined in this work? This should be explained.

Response: We add the magnetic particle inspection pictures corresponding to the accumulated traffic volume in Fig. 11 as a verification. Combined with the evaluation of DB experts, there are a large number of complex structural cracks and surface spalling in the 43Mt magnetic particle inspection results, and the crossing nose is in an early warning state at this time. However, the LSTM algorithm in this paper passes a sufficient number of yellow blocks that warn of failures at 33Mt. Sure enough, at 52Mt, there were large areas of falling blocks on the surface of the crossing nose, which seriously affected driving safety.

  1. How were the experimental data compared with calculation results? We did not see obvious experimental data correlated with fatigue damage.

Response: Thanks for the reviewer's question. I believe that the answer to question 5 can answer this question.

  1. It is suggested that "this method can be extended to all surface damages of rails" should be erased in the abstract since the evidence is not sufficient.

Response: Thanks to the reviewer's careful review, we have deleted it.

Additional comments:

1) The grammar of the sentence on lines 148-150 is wrong and it should be corrected. In addition, the word “etc.” should be deleted.

2) On line 401, the incorrect writings should be corrected. For example, “80\%” should be “80%.”

3) On line 407, “Of” should be “of.”

4) The grammar of the sentence on line 442 is wrong since there is lack of conjunctions between individual sentences.

5) Some irregular writings in the manuscript are required to be carefully examined and corrected. For example, on line 448, and line 456, the word “Figure” should be written in the form of “Fig.”

Response: Thanks to the reviewer's careful review, we have corrected the errors as requested.

Round 2

Reviewer 2 Report

The paper has been revised according to reviewers' suggestions.